# GMIW-Pose: Camera Pose Estimation via Global Matching and Iterative Weighted Eight-Point Algorithm

**Fan Chen, Yuting Wu, Tianjian Liao, Huiquan Zeng, Sujian Ouyang and Jiansheng Guan \***

School of Electrical Engineering and Automation, Xiamen University of Technology, Xiamen 361024, China
\* Correspondence: jsguan@xmut.edu.cn

**Abstract:** We propose a novel approach, GMIW-Pose, to estimate the relative camera poses between two views. This method leverages a Transformer-based global matching module to obtain robust 2D–2D dense correspondences, followed by iterative refinement of matching weights using ConvGRU. Ultimately, the camera's relative pose is determined through the weighted eight-point algorithm. Compared with the previous best two-view pose estimation method, GMIW-Pose reduced the Absolute Trajectory Error (ATE) by 24% on the TartanAir dataset; it achieved the best or second-best performance in multiple scenarios of the TUM-RGBD and KITTI datasets without fine-tuning, among which ATE decreased by 22% on the TUM-RGBD dataset.

**Keywords:** visual odometry; transformer; ConvGRU; the eight-point algorithm

## 1. Introduction

In recent years, the field of computer vision has made significant strides in various applications, including robotics, autonomous vehicles, and augmented reality. Camera pose estimation is a fundamental problem in computer vision, involving the determination of the relative position and orientation of two or more cameras observing the same scene, which plays a crucial role in 3D reconstruction, scene understanding, and localization. Accurate and robust camera pose estimation is essential for enabling machines to effectively perceive their surroundings and interact with them.

Traditional camera pose estimation methods are primarily based on geometric principles, using keypoint matching or optical flow between images to establish correspondences, followed by solving for camera poses using methods like epipolar geometry [1]. While these methods offer strong interpretability and often work well in many scenarios, they rely on accurate feature matching or optical flow, making them less robust in complex scenes. For example, in cases of texture absence, lighting variations, occlusion, dynamic objects, and so forth, feature matching or optical flow may result in errors or omissions, leading to pose estimation failures.

With the advancement of deep learning, some learning-based regression methods [2–4] have emerged. They predict the camera's pose directly from the RGB images using a pose network. These methods can adapt to complex scenes without the need for explicit feature extraction or point matching. However, according to the principles of multi-view geometry, recovering camera poses solely from monocular images encounters the challenge of scale ambiguity [5], meaning that a monocular image sequence cannot recover the scale of the scene. This implies that if deep learning models are used to regress camera poses directly without considering camera intrinsic parameters and geometric models, performance may degrade significantly when there are domain differences between the test and training datasets, such as different scenes, viewpoints, resolutions, and more. Therefore, the generalization ability of learning-based regression methods is often limited.

In this paper, we propose a novel approach called GMIW-Pose to estimate the relative camera pose between two views. It adheres to geometric model constraints while

utilizing a deep learning model to extract and enhance image features and optimize matching weights. An overall schematic is illustrated in Figure 1. GMIW-Pose comprises three key components:

1.  A global matching module based on the Transformer [6]: Inspired by LoFTR [7] and GMFlow [8], our approach employs a Transformer structure to enhance image features. It integrates global context features into local features through self-attention mechanisms and fuses features from two views using cross-attention mechanisms. The enhanced feature maps are used to generate a similarity matrix, from which dense correspondences between the two views are extracted via softmax operations. The Transformer efficiently captures long-range dependencies in image data to enhance local features for better key point matching. To balance performance and efficiency, our feature maps are at one-eighth of the original image size, resulting in coarse-level matches at the same scale.

2.  Robust camera pose estimation based on the weighted eight-point algorithm: The obtained coarse-level matches contain numerous outliers and noise. Common practice involves estimating an inlier set using methods like RANSAC [9] and then applying the eight-point or five-point algorithm [10] to recover poses. However, we found that RANSAC is not suitable in this context, as it assumes the presence of a certain number of inliers in the candidate set, while coarse-level matching results often have some bias. Using the weighted eight-point algorithm effectively mitigates the impact of outliers on the results, and it is differentiable, allowing end-to-end training of our model.

3.  Weight updating module with Convolutional Gated Recurrent Units (ConvGRU) [11]: We treat the problem of obtaining better-matching weights as an optimization problem, optimizing weights to minimize geometric loss. To achieve this, we introduce a weight-updating module that employs ConvGRU to simulate the optimization process. At each iteration, the ConvGRU module incorporates global image information, iterative context information, and geometric loss to iteratively refine the weights of correspondences.

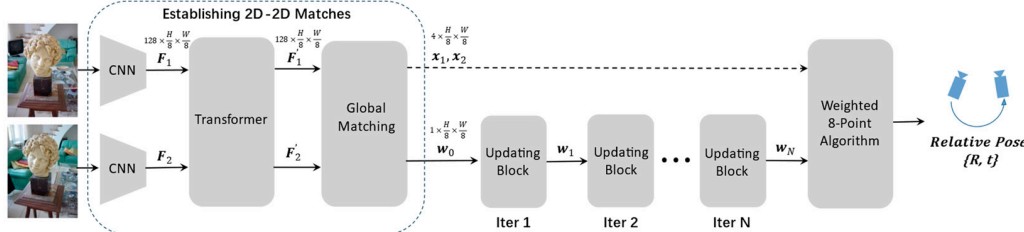

**Figure 1.** Overview of GMIW-Pose: The process begins with the extraction of downsampled dense features from two images using CNN and Transformer. Subsequently, through the global matching module, dense sets of matching points $\{x^A, x^B \in \mathbb{R}^{H \times W \times 2}\}$ between the two images and their corresponding initial weights $w_0 \in \mathbb{R}^{H \times W}$ are obtained. Following this, an updating module is employed to iteratively refine the weights. After N iterations, the weights $w_N$ along with $\{x^A, x^B\}$ can be used to compute camera's relative pose $\{R, t\}$ using the weighted eight-point algorithm.

The main contributions of this work are as follows: 1. We propose a novel algorithm framework, GMIW-Pose, to estimate the relative camera pose between two views. It uses global matching and the weighted eight-point algorithm to estimate camera poses, providing clear geometric interpretations and addressing the scale ambiguity challenge often encountered by pose regression-based methods. 2. We introduce the Transformer to enhance matching robustness, alleviating the poor matching performance issue in complex scenes faced by traditional methods. 3. We design a weight updating module based on ConvGRU to obtain better matching weights. To evaluate the performance of our proposed method, we conducted extensive experiments on the TartanAir [12] and KITTI [13] datasets.

The experimental results demonstrate that our GMIW-Pose outperforms existing methods in relative camera pose estimation.

## 2. Related Works

### 2.1. Two-View Camera Pose Estimation

Camera pose estimation is a thoroughly researched problem in the field of computer vision, with a wealth of methods available to address this challenging task. Classical approaches based on epipolar geometry have gained popularity due to their interpretability and generalization, being widely employed in various application domains and many advanced Structure from Motion (SfM) and Simultaneous Localization and Mapping (SLAM) systems [14–16]. The prerequisite for recovering geometric information is obtaining a set of sparse or dense matching points. Many classical algorithms have been used to establish matches, such as ORB [17], and SIFT [18], among others. Nevertheless, these traditional matching algorithms often perform poorly in cases of non-Lambertian surfaces, blurriness, and weak textures. Matching algorithms based on deep learning [7,19–21] have significantly alleviated these limitations. Among them, SuperGlue [21] utilizes graph neural networks (GNNs) to learn matching relationships from interest points and descriptors, while LoFTR [7] goes further by not only employing Transformers to capture long-range relationships within and between views but also bypassing the feature detection stage and directly producing dense matches. Subsequently, Transformers have been utilized for global optical flow estimation to address the long-standing challenge of large displacements [8,22]. Inspired by LoFTR and GMFlow, we also employ Transformers to enhance features and employ global matching methods to generate coarse-level dense matches.

On the other hand, learning-based methods for camera pose estimation have gained increasing attention in recent years. A significant category of these methods utilizes deep Convolutional Neural Networks (CNNs) [2–4] to directly regress camera poses, benefiting from their end-to-end nature, allowing them to predict camera poses directly from raw image pixels. These methods leverage deep neural networks to learn complex feature representations and efficiently estimate camera rotations and translations, effectively bypassing the costly feature matching or tracking steps. However, these models encounter generalization issues due to the presence of scale ambiguity in monocular vision [5] and typically require large amounts of annotated data for training, with high demands for data diversity and quality. To address the issue of generalization, some researchers have adopted unsupervised learning approaches, utilizing self-supervised signals for training. SfMLearner [23] minimizes the photometric loss between warped images and input images while simultaneously learning depth and pose. GeoNet [24] extends this idea to joint estimation of pose, depth, and optical flow. These methods typically learn camera poses by reconstructing loss from image sequences or multi-view scenarios, thus avoiding reliance on precise pose labels, although they still do not solve the scale ambiguity problem. Some works introduce geometric structures into deep learning models to circumvent scale issues. For instance, TartanVO [25] addresses the generalization problem by directly incorporating camera parameters into the model and training on a large amount of data. Jiang et al. [26] embed epipolar geometry constraints into a self-supervised learning framework through the joint optimization of camera poses and optical flow. In [27,28], they use the Eight-Point Algorithm as a neural network inductive bias to regress fundamental or essential matrices. Wang et al. [29] employ scale-invariant loss functions to train their model. Similarly, our approach also adopts a geometry-based structure, discarding PoseNet and using the Eight-Point Algorithm to compute camera poses. We employ a scale-invariant loss function to train the entire model end-to-end, further mitigating generalization issues caused by scale ambiguity.

### 2.2. Iterative Update

Earlier work [30–32] embedded optimization problems into network structures, often employing neural network models to predict the input or parameters of optimization

problems. BA-Net [33] incorporated a differentiable Levenberg–Marquardt (LM) algorithm into a neural network, using fully convolutional layers to predict the λ parameter at each update step of the LM algorithm. DeepV2D [34] iteratively updated motion and depth estimation, progressively converging to accurate numerical values. In recent developments in the field of optical flow estimation, many works [35–38] have employed iterative refinement to enhance optical flow. Among them, RAFT [38] utilized a Convolutional Gated Recurrent Unit (ConvGRU) module for iterative motion field updates to estimate optical flow. Inspired by this, in our work, we employ ConvGRU for optimization updates of matching weights. In each iterative step, ConvGRU takes global image information and geometric losses as inputs, predicts the residual weights, and progressively improves the weights of correspondences.

## 3. Method

For two RGB images with overlapping regions, denoted as $\{I^A, I^B \in \mathbb{R}^{H \times W \times 3}\}$, assuming that the camera's intrinsic parameters $K$ are known, the objective is to solve for the relative camera pose $\{R, t\}$. Here, $H$ represents the width of the images, and $W$ represents the height. $R \in SO(3)$ represents the camera's rotation, and $t \in \mathbb{R}^3$ represents the camera's translation. We propose a method based on global matching and an iterative weighted eight-point algorithm to compute the relative camera pose between the two images.

### 3.1. Establishing 2D–2D Matches

Inspired by LoFTR and GMFlow, we adopted a similar approach to extract 2D–2D dense matching point pairs between the two views. The difference lies in the trade-off between runtime speed and performance. We utilized only low-resolution coarse matching results for subsequent steps without refinement. Additionally, we computed not only forward optical flow but also backward optical flow and utilized the reciprocal of their sum to obtain the initial estimation of matching point pair weights, denoted as $w_0$.

### 3.1.1. Feature Extraction and Enhancement

First, we employed the Convolutional Neural Network ResNet-18 [39] to extract features from the original image $\{I^A, I^B\}$. To reduce computational complexity in subsequent steps, the resolution of the extracted feature maps $F^A$ and $F^B \in \mathbb{R}^{h \times w \times c}$ was reduced to $1/8$ of the original image size ($h = H/8, w = W/8$, where $c$ is the channel dimension of features). Next, we enhanced the extracted original features using a Transformer module. Specifically, we started by adding cosine positional embeddings to the original features, allowing them to carry position-related information. Subsequently, we iteratively applied self-attention mechanisms and cross-attention mechanisms to transform the feature maps. In self-attention layers, the query, key, and value all come from the same view's features. In cross-attention layers, the key and value are from the same view's features, while the query utilizes features from another view. This transformation process can be represented as:

$$F^A = \Gamma\left(I^A\right), F^B = \Gamma\left(I^B\right) \tag{1}$$

$$\widetilde{F}^A = \Theta\left(F^A + P\right), \widetilde{F}^B = \Theta\left(F^B + P\right) \tag{2}$$

Here, $\Gamma$ represents ResNet-18, $\Theta$ denotes the Transformer, $P$ is positional encoding and $\left\{\widetilde{F}^A, \widetilde{F}^B \in \mathbb{R}^{\frac{H}{8} \times \frac{W}{8} \times c}\right\}$ are enhanced feature maps. Positional encoding followed the 2D extension of the standard positional encoding as used in DETR [40].

Feature enhancement is a critical step in obtaining high-quality local features for matching. It blends global and position-related information into local features through self-

attention and cross-attention mechanisms, significantly improving matching performance in scenarios with weak textures, as illustrated in Figure 2.

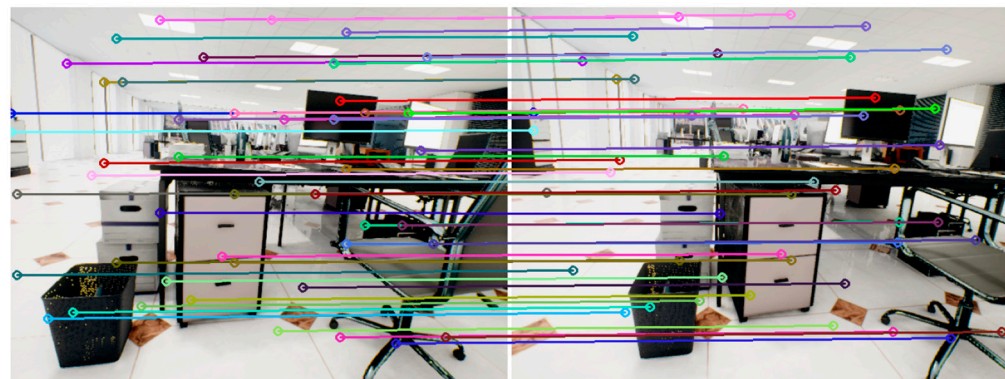

**Figure 2.** Matching Results Visualization. The figure displays a random subset of matching results, while the actual matching pairs are much denser. These colored lines connect corresponding matching points between the two images. It can be observed that even in regions with weak textures or repetitive patterns, such as ceilings and floors, good matching results are achieved.

### 3.1.2. Global Matching

The similarity of the features between two views can be measured by the inner product of corresponding feature vectors for each pixel. By flattening the spatial dimensions of $\overset{\sim}{F}{}^{A}$ and $\overset{\sim}{F}{}^{B}$ into two-dimensional matrices of size (h × w, c), we performed matrix multiplication, resulting in a large matrix of dimensions (h × w, h × w):

$$S = \frac{\langle \overset{\sim}{F}{}^{A}, \overset{\sim}{F}{}^{B} \rangle}{\sqrt{c}} \in \mathbb{R}^{h \times w \times h \times w} \tag{3}$$

Similar to the classical Transformer implementation, we divided by $\sqrt{c}$ to prevent excessively large inner products, ensuring stable gradients. The element in the i-th row and j-th column of $S$ represents the similarity between the i-th pixel in $I^A$ and the j-th pixel in $I^B$.

Various methods can be applied to extract dense matching point pairs between the two views from the similarity matrix. An approach is to search for the coordinates corresponding to the largest response in the similarity matrix $S$. Suppose the maximum value in the $S$ matrix's i-th row is located in the k-th column; then, the i-th pixel in $I^A$ is paired with the k-th pixel in $I^B$, yielding the matching point pair {i, k}. This process is repeated for each row, resulting in a total of $h \times w$ matching point pairs. While this method is straightforward and fast, it is non-differentiable and not suitable for stable training. LoFTR employed two optional solutions, the optimal transport layer and dual-softmax operator, to extract matches from $S$ in a differentiable manner, whereas GMFlow employed a single softmax for this operation. We adopted GMFlow's approach due to its efficiency and its ability to derive initial weights for the matching point pairs. Specifically, for all pixel coordinates $x^A \in \mathbb{R}^{h \times w \times 2}$ in image $I^A$, the corresponding matching points $x^B \in \mathbb{R}^{h \times w \times 2}$ in image $I^B$ were calculated as follows:

$$x^B = softmax_{row}(S)x^A \tag{4}$$

Here, $softmax_{row}$ denotes the softmax operation performed on each row of the $S$ matrix.

### 3.2. Weighted 8-Point Algorithm

In the case of known 2D–2D correspondences between two views and the extrinsic camera parameters $K$, we can calculate the fundamental matrix $F \in \mathbb{R}^{3 \times 3}$ and the essential matrix $E \in \mathbb{R}^{3 \times 3}$ between the two images using the normalized eight-point algo-

rithm. Subsequently, we can decompose the essential matrix to obtain the relative pose $\{R, t\}$. Specifically, according to the definition, the fundamental matrix $F$ satisfies the following equation:

$$X^B F\left(X^A\right)^T = 0 \tag{5}$$

where $X^A$ and $X^B$ are the three-dimensional homogeneous forms of pixel coordinates $x^A$ and $x^B$, respectively.

Next, by flattening the fundamental matrix $F$ into a 9-dimensional vector $f$ in row-major order, the above equation can be transformed into the form of a homogeneous linear system:

$$Af = 0 \tag{6}$$

Here, $A$ is the coefficient matrix obtained by arranging $X^A$ and $X^B$ according to a certain pattern. When the number of matched point pairs exceeds 8, this equation becomes an overdetermined system, which can be solved for the least squares solution of $f$ by using the singular value decomposition of $A^T A$. Additionally, since the rank of the fundamental matrix is 2, another SVD step is needed to enforce the determinant of $F$ to be 0.

The essential matrix can be derived from the fundamental matrix and camera extrinsics as follows:

$$E = K^T F K \tag{7}$$

Finally, the decomposition of $E$ can yield the translation $t$ and rotation $R$, based on the relationship between the essential matrix and relative pose:

$$E = [t]_\times R \tag{8}$$

where $[t]_\times$ denotes the skew-symmetric matrix corresponding to the three-dimensional vector $t$. Similarly, the decomposition algorithm can also utilize singular value decomposition.

In scenarios with noise or outliers in the correspondences, solving the fundamental matrix using the traditional eight-point algorithm may result in significant errors. A classical solution is to use techniques like RANSAC to estimate an inlier set and then compute the fundamental matrix using only the inliers. We employ a weighted least squares method to mitigate the influence of outliers, modifying Equation (6) as follows:

$$diag(w)Af = 0 \tag{9}$$

Here, $diag(w)$ represents a diagonal matrix generated from the weights $w$, where each element on the diagonal corresponds to the weight of a matched point pair. Consequently, the least squares solution for $f$ is obtained as the right singular vector corresponding to the smallest eigenvalue of the matrix $(diag(w)A)^T diag(w)A$. The subsequent steps follow the traditional approach to obtain the camera's pose $R$ and $t$.

### 3.3. Iterative Updates for Weights

3.3.1. Weights Initialization

For simplicity, the initial weights $w_0$ for dense matching pairs can be set to the identity matrix. However, to expedite convergence, we use the reciprocal of the sum of forward and backward optical flows as the initial values for weights. Forward optical flow can be further obtained from dense matching pairs $\left\{x^A, x^B\right\}$ as follows:

$$V^{AB} = x^B - x^A \in \mathbb{R}^{h \times w \times 2} \tag{10}$$

The backward optical flow $V^{BA} \in \mathbb{R}^{h \times w \times 2}$ can be obtained by swapping the positions of $I^A$ and $I^B$ in the above calculation. Here, due to the commutative properties of feature

extraction, feature enhancement, and similarity computation operations, the backward optical flow can be succinctly derived as follows:

$$V^{BA} = (softmax_{col}(S))^T x^A - x^A \tag{11}$$

If the matches are accurate, the absolute value of the sum of forward and backward optical flows $\Delta V = \left| V^{AB} + V^{BA} \right| \in \mathbb{R}^{h \times w \times 2}$ should be zero. However, due to incorrect matches, $\Delta V$ will be greater than zero, and the less accurate the matches, the larger the value of $\Delta V$. Therefore, the initial weights are set as follows:

$$w_0 = \frac{1}{\Delta V} = \frac{1}{\left| V^{AB} + V^{BA} \right|} \tag{12}$$

At this point, we have obtained dense matching pairs between the two views and their corresponding initial weights $\{ x^A, x^B, w_0 \}$.

### 3.3.2. Iterative Updates

To obtain improved matching weights, an update module is utilized to iteratively predict the residuals of weights at each step, with the core component being the ConvGRU unit. Let $w_k$ represent the matching weights optimized after $k$ iterations, and employing the weighted eight-point method allows us to estimate the corresponding essential matrix $\widetilde{F}_k$. A first-order approximation of geometric distance, namely the Sampson error [41], is employed as the error metric for the essential matrix. The matching weights $w$ are treated as variables to be optimized, enabling the minimization of the error function to progressively refine the estimate $\widetilde{F}$ towards the ground truth $F_{gt}$. For the Sampson error associated with the i-th pair of matching points in the k-th iteration, the specific calculation formula is as follows:

$$(e_i)_k = \frac{\left[ (X_i^B)^T \widetilde{F}_k X_i^A \right]^2}{\left( \widetilde{F}_k X_i^A \right)_x^2 + \left( \widetilde{F}_k X_i^A \right)_y^2 + \left( \widetilde{F}_k X_i^B \right)_x^2 + \left( \widetilde{F}_k X_i^B \right)_y^2} \tag{13}$$

Here, $X_i^A$ represents the three-dimensional homogeneous coordinates of the i-th pixel in image $I^A$, and $\left( \widetilde{F}_k X_i^A \right)_x^2$ denotes the square of the x-coordinate value of the three-dimensional vector $\widetilde{F}_k X_i^A$.

Next, we employed the ConvGRU unit for optimization. The encoding results of $w_k$ and $e_k$, along with the context $c$, are concatenated as the $Input_k$ to obtain the next hidden state $H_{k+1}$:

$$Input_k = concat[\Pi(w_k, e_k), c] \tag{14}$$

$$H_{k+1} = ConvGRU(Input_k, H_k) \tag{15}$$

Here, $\Pi$ is a Convolutional Neural Network used to encode the weights $w_k$ and errors $e_k$. The context $c$ and the initial hidden state $H_0$ are obtained by linear transformations of the enhanced reference image feature $F^A$, followed by channel-wise splitting.

The weight residual $\Delta w_k$ at the k-th step is obtained from $H_{k+1}$ through an output head, which is also a three-layer Convolutional Neural Network. The update equations are as follows:

$$\Delta w_{k+1} = weight_{head(H_{k+1})} \tag{16}$$

$$w_{k+1} = w_k + \Delta w_{k+1} \tag{17}$$

### 3.4. Training Loss

We supervised the 2D–2D consistency prediction using the absolute difference between the forward optical flow $V^{AB}$ and the ground truth optical flow $V_{gt}^{AB}$:

$$L_{match} = \left\| V^{AB} - V_{gt}^{AB} \right\|_1 \tag{18}$$

The pose loss is divided into two parts: rotation loss and translation loss:

$$L = \sum_{k=0}^{N-1} \gamma^{N-1-k} \left( L_k^{rot} + \beta L_k^{trans} \right) \tag{19}$$

Here, $N$ represents the total number of iterations for updates. $\gamma$ (set to 0.8) is used to adjust the weight of the loss function corresponding to each iteration, with higher weights assigned to later updates. $L_k^{rot}$ and $L_k^{trans}$ represent the rotation and translation losses after the k-th step of updates, respectively. $\beta$ is a factor used to balance the scales of these two losses, and in this paper, $\beta$ is set to 1. The definitions of $L_k^{rot}$ and $L_k^{trans}$ are as follows:

$$L_k^{rot} = \min \left( \frac{3 - tr\left( R_{gt}^T \widetilde{R}_k \right)}{4}, \tau_{rot} \right) \tag{20}$$

$$L_k^{trans} = \min \left( \frac{1 - \frac{\widetilde{t}_k \cdot t_{gt}}{\left\| \widetilde{t}_k \cdot t_{gt} \right\|}}{2}, \tau_{trans} \right) \tag{21}$$

Here, $\widetilde{R}_k$ and $\widetilde{t}_k$ represent the predicted rotation matrix and translation vector after the k-th step of updates, while $R_{gt}$ and $t_{gt}$ represent their ground truth counterparts. $tr(*)$ denotes the trace of a matrix, and $\| * \|$ represents the magnitude of a vector. $\tau_{rot}$ (set to 0.0001) and $\tau_{trans}$ (set to 0.1) are used to limit the upper bounds of the loss functions, ensuring training stability. It is worth noting that the translation loss is a scale-independent function, depending only on the angle between predicted and ground truth translations. This choice is made because monocular reconstruction can recover scene structure but not its scale, and using a scale-independent loss function improves the model's generalization capability.

## 4. Experiments and Results

### 4.1. Datasets

We conducted training exclusively on the training dataset from TartanAir [12], which includes the "Hard" portions of all 18 scenes. In addition, we used eight sequences MH000-MH007 from the TartanAir challenge dataset as a validation set to evaluate the performance of our method. Since TartanAir is a synthetic dataset, to demonstrate the effectiveness of our algorithm in real-world scenarios, we conducted validation on the KITTI dataset [13] without pretraining on this dataset.

### 4.2. Implementation Details

Our model was implemented using PyTorch [42] and Kornia [43] libraries. Kornia is a differentiable computer vision library built on PyTorch, and we utilized it for fundamental matrix estimation and differentiable decomposition of the essential matrix. We utilized the AdamW optimizer and clipped the network gradients to ensure that their L2 norm is limited to 1. The initial learning rate was set to 0.0001, and we used PyTorch's ReduceLROnPlateau as a learning rate scheduler, with patience set to 1 and a factor of 0.2. To strike a balance between performance and efficiency, we set the total number of iterations for updates $N$ to

2. During training, we used a batch size of 16 on the TartanAir dataset, initially training the feature extraction network Γ and the feature enhancement network Θ with the matching loss function for 50,000 steps. Subsequently, we performed end-to-end training of the entire model using the pose loss function for an additional 50,000 steps.

*4.3. Evaluation*

Similar to prior research, we used the Absolute Trajectory Error (ATE) [44] on the TartanAir challenge dataset to measure the performance differences between different methods. ATE represents the absolute difference between the ground truth and estimated poses, providing an intuitive reflection of the global consistency between the predicted and actual trajectories. To align the coordinate systems and scales of the two trajectories, we needed to compute a similarity transformation matrix $S \in Sim(3)$ that maps estimated poses to ground truth poses. The definition of ATE is as follows:

$$ATE = \left( \frac{1}{m} \sum_{t=1}^{m} \left\| trans\left( Q_t^{-1} S P_t \right) \right\|^2 \right)^{\frac{1}{2}} \tag{22}$$

Here, $P_t \in SE(3)$ and $Q_t \in SE(3)$ are the estimated and ground truth poses corresponding to the t-th frame of the trajectory, respectively. $trans(\cdot)$ denotes the translation part of a pose matrix.

We used the sequences MH000–MH007 from the TartanAir dataset as a validation set and measured the ATE metric to evaluate the model's performance. The experimental results are shown in Table 1. Compared to the ORB-SLAM, TartanVO, and DiffPoseNet methods, our approach achieved the best results in six out of the eight sequences (MH000–MH007) and had the lowest overall average ATE, reducing it by 24% compared to the second-best method. Furthermore, on the MH004 and MH005 datasets, the traditional method ORB-SLAM failed to track and produce results, indicating that learning-based methods exhibit stronger robustness compared to traditional methods.

**Table 1.** ATE(m)↓ on the MH sequences of the TartanAir dataset. Bold text represents the best results, while underlined text represents the second-best results.

| Methods | MH000 | MH001 | MH002 | MH003 | MH004 | MH005 | MH006 | MH007 | Average |
|---|---|---|---|---|---|---|---|---|---|
| ORB-SLAM [14] | <u>1.30</u> | **0.04** | 2.37 | 2.45 | - | - | 21.47 | 2.73 | - |
| TartanVO [25] | 4.88 | 0.26 | 2.00 | 0.94 | <u>1.07</u> | 3.19 | <u>1.00</u> | 2.04 | 1.92 |
| DiffPoseNet [45] | 2.56 | 0.31 | <u>1.57</u> | <u>0.72</u> | **0.82** | <u>1.83</u> | 1.32 | **1.24** | <u>1.30</u> |
| Ours | **1.24** | <u>0.15</u> | **0.67** | **0.29** | 1.50 | **1.43** | 0.89 | **1.24** | **0.93** |

To demonstrate the effectiveness of GMIW-Pose on real datasets, we validated our method on five sequences of the TUM-RGBD dataset: 360, desk, desk2, rpy, and xyz. TUM-RGBD is a real dataset captured using handheld devices, covering multiple scenarios such as offices, corridors, and lobbies. As shown in Table 2, our method achieved the best performance in three of these scenarios, with the overall average Absolute Trajectory Error (ATE) being the smallest, which is 22% lower than the second-best method.

**Table 2.** ATE(m)↓ on TUM-RGBD dataset. Bold text represents the best results, while underlined text represents the second-best results.

| Methods | 360 | desk | desk2 | rpy | xyz | Average |
|---|---|---|---|---|---|---|
| ORB-SLAM2 [46] | - | **0.016** | 0.078 | - | **0.004** | - |
| TartanVO [25] | 0.178 | 0.125 | 0.122 | 0.049 | 0.062 | 0.107 |
| DiffPoseNet [45] | <u>0.121</u> | 0.101 | <u>0.053</u> | <u>0.056</u> | 0.048 | <u>0.076</u> |
| Ours | **0.109** | <u>0.057</u> | **0.042** | **0.045** | <u>0.040</u> | **0.059** |

In addition, we conducted experiments on four sequences (06, 07, 09, and 10) from the KITTI tracking dataset. The KITTI dataset contains real-world traffic scene data that are well-calibrated and the dataset is commonly used as a benchmark in the SLAM field. We used the KITTI metric [13] as the evaluation criteria, which calculates the average drift distance ($t_{rel}$ in m/100 m) and rotation error ($R_{rel}$ in degree/m) for every 100 m of trajectory in the range of 100 to 800 m. We compared our method with other approaches, including DeepVO, Wang et al., UnDeepVO, GeoNet, TartanVO, BiLevelOpt, ORB-SLAM, VISO2-M, and DiffPoseNet, and the experimental results are presented in Table 3. The results show that our method achieved either first or second place in the majority of the test sequences. The average drift distance decreased by 8% compared to DiffPoseNet. It is worth noting that our model, like TartanVO and DiffPoseNet, was trained only on the TartanAir dataset and was not fine-tuned on the KITTI dataset. This not only demonstrates the effectiveness of GMIW-Pose on real-world datasets but also highlights its strong generalization capabilities.

**Table 3.** $t_{rel}\downarrow$ and $R_{rel}\downarrow$ on KITTI dataset. Bold text represents the best results, while underlined text represents the second-best results.

| Methods | 06 | | 07 | | 08 | | 09 | | 10 | | Average | |
|---|---|---|---|---|---|---|---|---|---|---|---|---|
| | $t_{rel}$ | $R_{rel}$ | $t_{rel}$ | $R_{rel}$ | $t_{rel}$ | $R_{rel}$ | $t_{rel}$ | $R_{rel}$ | $t_{rel}$ | $R_{rel}$ | $t_{rel}$ | $R_{rel}$ |
| DeepVO [47] | 5.42 | 5.82 | 3.91 | 4.60 | - | - | 8.11 | 8.83 | - | - | - | - |
| Wang et al. [48] | - | - | - | - | 8.04 | 1.51 | 6.23 | 0.97 | - | - | - | - |
| UnDeepVO [49] | 6.20 | 1.98 | **3.15** | 2.48 | - | - | 10.63 | 4.65 | - | - | - | - |
| GeoNet [24] | 9.28 | 4.34 | 8.27 | 5.93 | 26.93 | 9.54 | 20.73 | 9.04 | 16.30 | 7.21 | 16.30 | 7.21 |
| TartanVO [25] | 4.72 | 2.95 | 4.32 | 3.41 | 6.03 | 3.11 | 6.89 | 2.73 | 5.49 | 3.05 | 5.49 | 3.05 |
| BiLevelOpt [26] | - | - | - | - | 4.36 | 0.69 | 4.04 | 1.37 | - | - | - | - |
| ORB-SLAM [14] | 18.68 | **0.26** | 10.96 | **0.37** | 15.3 | **0.26** | <u>3.71</u> | **0.3** | 12.16 | **0.30** | 12.16 | **0.30** |
| VISO2 m [50] | 7.34 | 6.14 | 23.61 | 19.11 | 4.04 | 1.43 | 25.2 | 3.84 | 15.05 | 7.63 | 15.05 | 7.63 |
| DiffPoseNet [45] | <u>2.94</u> | 1.76 | 4.06 | 2.35 | <u>4.02</u> | <u>0.51</u> | 3.95 | 1.23 | <u>3.74</u> | <u>1.46</u> | <u>3.74</u> | 1.46 |
| Ours | **2.59** | <u>1.39</u> | <u>3.89</u> | <u>2.13</u> | **3.55** | 1.45 | **3.68** | <u>0.88</u> | **3.43** | <u>1.46</u> | **3.43** | <u>1.41</u> |

### 4.4. Ablations

- **Robust Estimation of the Fundamental Matrix:** In GMIW-Pose, we initially used the weighted eight-point algorithm to estimate the fundamental matrix, but we replaced it with RANSAC and LMedS for comparison, as shown in Table 4. The experimental results indicate a significant performance drop when replacing the weighted eight-point algorithm with RANSAC and LMedS [51]. This drop in performance can be attributed to the fact that the 2D–2D matching at the front end is performed on coarse-level features, leading to bias in the set of inlier matches selected by RANSAC and LMedS. In contrast, the weighted eight-point algorithm not only effectively addresses this situation but is also differentiable, allowing for a smooth training process.

**Table 4.** ATE(m)$\downarrow$ on the MH sequences of the TartanAir dataset for the variants of GMIW-Pose. Bold text represents the best results, while underlined text represents the second-best results.

| Methods | MH000 | MH001 | MH002 | MH003 | MH004 | MH005 | MH006 | MH007 | Average |
|---|---|---|---|---|---|---|---|---|---|
| RANSAC [9] | 14.11 | 2.24 | 7.26 | <u>1.27</u> | <u>2.20</u> | 15.27 | 4.40 | <u>7.22</u> | 6.75 |
| LMedS [51] | <u>12.60</u> | <u>1.19</u> | <u>6.24</u> | 1.73 | 2.39 | <u>6.80</u> | <u>1.04</u> | 13.45 | <u>5.68</u> |
| Ours | **1.24** | **0.15** | **0.67** | **0.29** | **1.50** | **1.43** | **0.89** | **1.24** | **0.93** |

- **Number of Iterations:** We compared the impact of different numbers of iterations on performance, as illustrated in Figure 3. As the number of iterations N in the update module increases, the ATE error decreases. However, beyond N = 2, increasing the number of iterations has diminishing returns on model performance. Considering the

balance between computational efficiency and performance, setting N to 2 is a suitable choice for practical use.

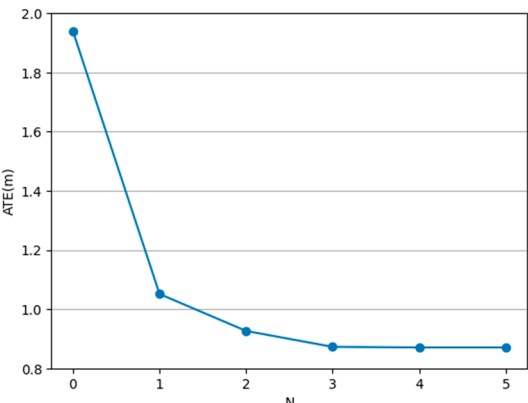

**Figure 3.** ATE(m)↓ on the MH sequences of the TartanAir dataset for different numbers of iterations N.

## 5. Conclusions

In this paper, we propose a novel method, GMIW-Pose, to estimate camera relative poses. To address the generalization issues faced by learning-based direct methods due to scale ambiguity, we adopt a structure based on epipolar geometry, which offers strong interpretability. Additionally, we introduce feature enhancement and global matching to establish dense consistency between two views, mitigating the limitations of traditional matching methods in complex scenes. Finally, to robustly estimate camera poses from dense consistency, we employ the weighted eight-point algorithm and iteratively optimize matching weights using the ConvGRU module. The experimental results show that GMIW-Pose achieved advanced performance on the TartanAir dataset and can be generalized to the TUM-RGBD and KITTI real datasets without training.

Furthermore, the weighted dense matching pairs obtained by GMIW-Pose can be directly used to reconstruct dense three-dimensional scenes. In the future, we plan to extend its application to multi-view scenarios, leveraging information from multiple frames for further optimization to achieve superior performance. Specifically, we can extend the current two-view epipolar geometry constraints to three-view trifocal tensor constraints and even more views [52]. In addition to geometric constraints, photometric consistency among multiple views is also a crucial constraint. Differentiable rendering techniques [53,54] can be employed to reconstruct images, minimizing photometric losses between the original images and the reconstructed images and facilitating self-supervised pose learning.

**Author Contributions:** Conceptualization, F.C. and J.G.; methodology, F.C. and J.G.; software, F.C.; validation, F.C., Y.W., T.L., H.Z. and S.O.; writing, F.C.; review and editing, J.G., Y.W., T.L., H.Z. and S.O.; project administration, J.G. All authors have read and agreed to the published version of the manuscript.

**Funding:** This work was supported by the Key Project of the Natural Science Foundation of Fujian Province No.2020J02048, and Xiamen Municipal Natural Science Foundation 3502Z20227215, and Xiamen Ocean and Fisheries Development Special Fund Youth Science and Technology Innovation Project 23ZHZB043QCB37.

**Data Availability Statement:** Publicly available datasets were analyzed in this study. This data can be found here: https://learn.microsoft.com/azure/open-datasets/dataset-tartanair-simulation, accessed on 13 December 2022.

**Conflicts of Interest:** The authors declare no conflict of interest.

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
