# Peer review of "GMIW-Pose: Camera Pose Estimation via Global Matching and Iterative Weighted Eight-Point Algorithm"

_electronics, doi:10.3390/electronics12224689_

Round 1
Reviewer 1 Report
Comments and Suggestions for Authors
The article is very interesting and concerns the assessment of the change in camera position based on differences in individual images. The content was presented in a transparent and logical manner. Only drawing no. 1 needs improvement. Its resolution is very low, so it is quite difficult to read. An interesting next step of the research would be to show the consistency of the mapping of known camera movement trajectories based on the analysis of images obtained from it.
Reviewer 2 Report
Comments and Suggestions for Authors
The paper is well-written and well-organized, and the importance of the paper is clear to me.
The main contribution of the paper is combining CNN and Transformer with a matching algorithm for the camera pose estimation problem.
I only have two comments/recommendations:
Comment 1:
For the camera pose estimation problem, the pose matrix $(R, t)$ is given in Section 3. Where $R \in SO(3)$, and $t \in \R^3$. Observer that $R \in SO(3)$ does not lie in the Euclidean spaces. Therefore, in your learning algorithm, you should consider the special orthogonal space of the rotation matrix. How did you solve the non-smoothness problems? Can you use quaternions instead of $R \in SO(3)$?
Comment 2:
One of the most critical challenging parts of the pose estimation is the feasibility of the optimization algorithms under the inherited constraints of the problem. For example, if you want to use inter-frame information, you need some challenging constraints. How can you consider the hard constraints in your framework? Is it possible? Presumably, it would be best if you modified the loss (objective) function. There are only a few papers about the calibration problems related to pose estimation that consider hard constraints in their optimization framework:
https://doi.org/10.1177/0278364919844824
https://doi.org/10.1109/LRA.2022.3180428
https://doi.org/10.1109/ICRA48891.2023.10161070
I believe that this paper will be beneficial for the community. Therefore, if you can explain some possible extensions of the work in the future work section, it would be better for the community.
Reviewer 3 Report
Comments and Suggestions for Authors
This paper describes a deep learning based method to estimate the relative pose between two views. The authors propose a transformer based global matching method to obtain dense correspondences between the two views. This is followed by an iterative refinement procedure involving again a CNN ConvGRU.
Major comments:
1. The conclusion that the proposed method is superior for real datasets needs more evidence from experiments. If I look at table 2, R_{rel} in all cases is superior to the proposed method, whereas t_{rel} is better for the proposed method. The authors have not attemped to explain this trend. I would strongly suggest the authors to please explain why R_{rel} is inferior for all cases? Even if a reasonable explanation is given, the authors cannot state that their method is superior. I would strongly suggest to run more experiments on other datasets if available to figure out the issue and improve on it.
Minor comments:
1. Line 333-336 can be made into a table or at least shortened somehow.
2. Line 346 does not make sense. How can the range me a singular number?
3. Line 322, which norm is used?
Round 2
Reviewer 3 Report
Comments and Suggestions for Authors
The authors have addressed all my concerns and incorporated my suggestions. Thanks for clarifying the difference between R and t scoring.